# Variation in the Early Life and Adult Intestinal Microbiome of Intra-Uterine Growth Restricted Rat Offspring Exposed to a High Fat and Fructose Diet

**DOI:** 10.3390/nu15010217

**Published:** 2023-01-01

**Authors:** Liesbeth V. Maggiotto, Shubhamoy Ghosh, Bo-Chul Shin, Amit Ganguly, Venu Lagishetty, Jonathan P. Jacobs, Sherin U. Devaskar

**Affiliations:** 1Department of Pediatrics, Division of Neonatology & Developmental Biology and UCLA Children’s Discovery and Innovation Institute, David Geffen School of Medicine at UCLA, Los Angeles, CA 90095, USA; 2The Vatche and Tamar Manoukian Division of Digestive Diseases, David Geffen School of Medicine at UCLA, Los Angeles, CA 90095, USA; 3Division of Gastroenterology, Hepatology and Parenteral Nutrition, Veterans Affairs Greater Los Angeles Healthcare System, Los Angeles, CA 90073, USA

**Keywords:** microbiota, gut, IUGR, Western diet

## Abstract

Intra-Uterine Growth Restriction (IUGR) is a risk factor for many adult-onset chronic diseases, such as diabetes and obesity. These diseases are associated with intestinal microbiome perturbations (dysbiosis). The establishment of an intestinal microbiome begins in utero and continues postnatally (PN). Hypercaloric diet-induced dysbiosis is a major driver of childhood obesity. We hypothesized that different postnatal diets superimposed on IUGR will alter the postnatal intestinal microbiome. We compared four experimental rat groups: (1) Ad lib fed regular chow diet pre- and postnatally (CON), (2–3) IUGR induced by maternal caloric restriction prenatally followed postnatally (PN) by either (2) the control diet (IUGR-RC) or (3) High-Fat-high-fructose (IUGR-HFhf) diet, and lastly (4) HFhf ad lib pre- and postnatally (HFhf). Fecal samples were collected from dams and male and female rat offspring at postnatal day 2, 21, and adult day 180 for 16S rRNA gene sequencing. Maternal diet induced IUGR led to dysbiosis of the intestinal microbiome at PN21. Postnatal HFhf diet significantly reduced microbial diversity and worsened dysbiosis reflected by an increased Gammaproteobacteria/Clostridia ratio. Dysbiosis arising from a mismatch between IUGR and a postnatal HFhf diet may contribute to increased risk of the IUGR offspring for subsequent detrimental health problems.

## 1. Introduction

Gut microbiota plays a key role in host digestion and metabolism. Composition of gut microbiota is highly variable and can be influenced by different factors including diet. Gut microbiomes are dominated by a few major phyla, i.e., Firmicutes, Bacteroidetes, Actinobacteria, Proteobacteria, Fusobacteria, and Verrucomicrobia, with the two phyla Firmicutes and Bacteroidetes representing 90% of gut microbiota [1].

There is growing evidence that many chronic diseases have their origin in fetal life. Intra-uterine growth restriction (IUGR) is a major risk factor for chronic diseases in later life. These IUGR infants are at increased risk for developing metabolic syndrome, diabetes mellitus, cardiovascular disease and obesity [2]. Particularly, infants that are exposed to a mismatch between prenatal and postnatal nutritional environments are most vulnerable to developing non-communicable chronic diseases. This has been referred to as the “Mismatch Concept” [3,4].

Recent investigations support the formulation of a dysbiotic intestinal microbiome early in life playing a role in the development of chronic disorders such as childhood obesity and autoimmune disorders [5]. Childhood obesity is one of the most serious public health challenges of the 21st century. About 1 in 5 children in the U.S. is obese, putting them at increased risk for diabetes and cardiovascular disease. A Western diet with high fat and high fructose content is one of the major drivers of childhood obesity. Diet plays a large role in the phylogenetic makeup of the microbiome [6,7,8,9,10]. Past research showed decreased *Bifidobacterium* in infancy to be associated with an increased risk for being overweight at 10 years of age. Other research reported that when microbiota is transplanted from genetically obese mice into germ free normal mice, it caused the recipient mice to become obese—despite being maintained on the same diet and the same amount of chow [11,12]. 

While investigations in newborn IUGR pigs [13,14] and very low birth weight premature infants [15] have been conducted at one time point that suggests dysbiosis, to date there are no studies relating the influence of prenatal and/or postnatal diet upon the life course of the IUGR microbiome. To determine the role of the postnatal microbiome of an IUGR offspring along with the influence of prenatal and/or postnatal oral diet upon the postnatal and adult outcomes, we conducted the present investigation. We hypothesized that the consumption of a hypercaloric diet overlaid upon an IUGR phenotype thereby creating a prenatal/postnatal nutritional mismatch, will lend itself to a dysbiotic intestinal microbiome, which in turn may be responsible for the immediate postnatal and ultimate adult phenotype previously reported. This phenotype consists of obesity and glucose intolerance encountered in the adult stage of life [16,17].

To test this hypothesis, we undertook the present study employing a previously validated prenatal calorie restricted rat IUGR model [16,17] and examined both the male and female offspring during light and dark phases as well.

## 2. Materials and Methods

### 2.1. Methods

#### Animals 

The study was approved by the Animal Research Committee of the University of California Los Angeles protocol# 1999-104-61. Pregnant Sprague Dawley rats (*n* = 48 total) (Charles River Laboratories, Hollister, CA, USA) were delivered on gestational day 2 and housed at the UCLA Clinical Health Sciences Building under the supervision of DLAM. Each pregnant rat was housed in individual cages at 21–23 °C within 12 h light/dark cycles. Cages were autoclaved and the cage bedding was sterile. The rats had free access to water and regular chow ad libitum (Teklad NIH-31-7013, carbohydrates 59%, protein 23% and fat 18% from kilocalories) (Envigo, Madison, WI, USA). 

### 2.2. Study Design

At gestational day 11 (G11) dams were randomized to either continue regular chow (CON group) or they were placed on a Western diet ad libitum that was high in fat and fructose (HFhf group) (Harlan-Teklad TD.88137, carbohydrates 43%, protein 15%, and fat 42% from kilocalories) along with 25% fructose in their drinking water which was accessible ad libitum both during prenatal and lactational/postnatal stages. The pregnant dams in the IUGR group were 50% calorie restricted beginning from gestation day (G) 10 in their access to regular chow diet until the birth of their pups. Soon after birth, the maternal diet was switched to either regular chow accessed ad libitum forming the IUGR-RC (regular chow) group, or a HFhf diet creating the IUGR-HFhf group (Figure 1).

At birth, the litter size/dam was culled to 8 pups/litter maintaining 4–10 litters/experimental group. The remaining pups were examined at postnatal (PN) 2, at the beginning of the suckling period. At PN21 all pups and dams were sacrificed at 2 time points 12 h apart (Zeitgeber [ZT] 4 and ZT16) and stool samples were collected separately from males and females at both PN2 and PN21. The gut microbiome studies at PN21 was timed to occur at the end of their suckling period. In addition, some litters were allowed to continue until the adult stage of 180d, at which point their gut microbiome was also examined.

### 2.3. Anthropometric Measurements and Sample Collection

All dams were weighed weekly, and the pups were weighed every 2–3 days until PN21 [18]. 

We surgically collected fecal samples from within the intestines of PN2 and PN21 pups when euthanized. At dissection of PN2 pups, the colorectal tissues were flushed with 500 µL of sterile water and fecal samples were collected in sterile Eppendorf tubes. At PN21 it was possible to directly scrape out the dry fecal samples which were then stored in sterile tubes. Maternal (gestational day 19–20) and the 180d adult offspring stool samples were collected into sterile tubes, along with the stool samples of the offspring at PN2 and PN21, using gloves and autoclaved equipment. All collected fecal samples were stored at −80 °C until further analyses. 

### 2.4. 16S rDNA Microbiota Profiling

Fecal samples that were freshly collected from pregnant mothers, postnatal or adult offspring were subjected to bacterial genomic DNA extraction using the Qiagen Powersoil kit. Libraries were generated according to methods adapted from Caporaso et al. [19] as previously described [20]. The V4 region of the 16S rRNA gene was amplified with individually barcoded universal primers. PCR amplified products were purified and pooled in equimolar concentrations and sequenced using the Illumina MiSeq platform and 2 × 250 bp reagent kit for paired-end sequencing.

### 2.5. Sequencing Data Processing and Analysis

Demultiplexed sequence data were imported to QIIME2 [21] and quality filtering was performed using DADA2 [22] to ensure adequate sequence quality. A total of 6,892,903 number of reads were obtained from PN2 and PN21 samples with an average of 49,200 reads per sample while a total of 10,271,286 reads were gathered from the adult samples (maternal and 180d old adult offspring) with an average of ~98,000 reads per sample. We observed that the error distributions were different for the postnatal and adult rat microbiome datasets, hence we analyzed them separately. A feature table along with a representative sequence table containing amplicon sequence variants (ASV) were generated for each dataset. ASVs with low counts across all the samples were removed. Following denoising, a rooted phylogenetic tree was constructed based on sequence similarity to infer the evolutionary relationship between microorganisms. The outputs from QIIME2 (feature table, taxonomic classifications, phylogenetic tree) along with the corresponding metadata, were imported to R (version 4.2.1) and assembled into a phyloseq object using the phyloseq package (version 1.40.0) [23]. To estimate alpha diversity, Shannon, Inverse Simpson, and Faith’s phylogenetic alpha diversity indices were calculated for all four dietary groups using R package mia [24]. The Bray–Curtis distance metric was used to assess compositional patterns (beta diversity) at ASV levels. Differential abundances (DA) of bacterial taxa at the phylum and genus levels were calculated with the bias correction test using ANCOM-BC [25].

### 2.6. Data Visualization 

Alpha diversity data generated in QIIME2 were imported to R, where alpha diversity box plots were generated using the ggplot2 package (version 3.3.3). Beta diversity results visualized by principal coordinates analysis (PCoA) were generated in QIIME2 and exported to R with the Qiime2R package (version 0.99.5). For differential abundance analysis between dominant diet groups, the log2 fold change of microorganism taxa at the class level was plotted using ggplot2 as well.

Statistical Analysis:

Body weights were compared in an age-matched and sex-matched manner between the four groups employing the ANOVA followed by the Tukey post hoc test towards detecting inter-group differences. Kruskal–Wallis test was employed to detect differences in the microbial α-diversity indices across the four experimental groups, followed by the post hoc Dunn’s test to detect inter-group differences. PERMANOVA was used to analyze the Bray–Curtis dissimilarity and the principal coordinate analysis (PCoA) of the microbial β-diversity analyses. The bias correction method of ANCOM-BC was employed to calculate differential abundances (DA) of the bacterial taxa at the phylum and genus levels (package from R).

## 3. Results

Both IUGR offspring and HFhf pups were significantly smaller than the control pups. The IUGR litters were the largest- almost 13 pups per litter, compared to 10 for the HFhf litters (Appendix A). When correcting the pups’ weight for their litter size, the HFhf pups were the smallest followed by the IUGR pups. There were significantly more females (59%) in the IUGR litters compared to CON (47%) and HFhf (39%) groups. At birth in both males and females the IUGR offspring and HFhf pups were significantly lighter than CON pups. More specifically, the IUGR-HFhf pups were significantly smaller at most time points for both males and females. Only the IUGR-RC males after PN16 displayed catch up growth, which was not evident in the females within the first 21 days of life. In fact, there were significant sex differences only in the IUGR-RC group (Figure 2).

### 3.1. Gut Bacterial Diversity and Composition across Ages and Dietary Compositions

Processing with DADA2 identified 919 amplicon sequence variants (ASVs) from PN2 and PN21 samples and around 2300 ASVs from both types of adult samples after removing ASVs with low counts across all the samples. Differences in bacterial composition among various dietary groups at different ages were assessed at the Phylum and Genus levels. Seven different phyla were detected with more than 2% mean abundance across the ages and different groups (Figure 3A). Phylum Proteobacteria appeared to be predominant among the PN2 animals of all four experimental groups. Only the female PN2 offspring from the IUGR group demonstrated an abundance of phyla Actinobacteria, but with no representation of Bacteroidetes. Whereas at PN21 all four experimental groups were predominantly represented by three phyla (Bacteroidetes, Firmicutes and Verrucomicrobia). The presence of Deferribacteres was evident only in the CON group offspring (Figure 3A). ~77 bacterial genera were detected at PN2 and PN21 groups compared to 54 genera between the two sets of adult groups (maternal and adult offspring). At genus level, we saw high abundance of *Rothia* at PN2, while PN21 and the adults had high abundance of *Roseburia* and *Prevotella*, respectively (Figure 3B). 

### 3.2. Effects of Dietary Influence on Gut Bacterial Diversity at Postnatal Stages PN2 and PN21

To assess the effects of various dietary exposures on gut microbial diversity at PN2, we calculated alpha and beta diversity employing a variety of scoring indexes for both forms of diversity. Alpha diversity is a measure of species diversity in a particular niche, in our study this refers to the gut lumen of PN2 mouse offspring from the different experimental groups. To this end, we have assessed alpha diversity (species richness), Shannon (species richness and diversity), Simpson (species evenness) and Phylogenetic diversity (PD). At PN2, no significant difference in alpha diversity was observed except in the IUGR group that showed higher PD compared to the CON group (Figure 4A–D, Appendix A). No significant differences were attributable to the sex of the PN2 offspring. Significant differences were observed at PN2 in bacterial composition between the two groups (IUGR vs. CON) when using Bray–Curtis dissimilarity (*p* = 0.036, PERMANOVA) (Figure 4E). Significant differences in bacterial composition that emerged at PN2 with post hoc analysis of composition with bias correction (ANCOM-BC) testing detected two Phyla—Verrucomicrobia and Bacteroidetes—that were increased in IUGR versus CON. At the genus level, *Akkermansia*, *Parabacteroides, Fusobacterium, Agrregatibacter, Campylobacter* and *Clostridium Neisseriaceae* were increased, while *Leptotrichia,* and *Morganella* were reduced significantly in the IUGR versus CON. Concurrently, we found a significant decrease of Phyla Fusobacteria in the PN2 HFhf group. We also observed significant increases in the *Enterococcus* and *Parabacteroides* genera and decreases in *Lautropia* and *Morganella* (Figure 4F,G) in the PN2 HFhf group when compared to CON.

Further investigation at PN21 revealed significantly lower alpha diversity in the high fat high fructose fed groups, namely HFhf alone or IUGR-HFhf. However, no significant change in alpha diversity was evident in the IUGR-RC group when compared to CON (Figure 5A–D). Significant differences (*p* = 0.001, PERMANOVA) in bacterial composition were observed between late postnatal samples (PN21) using Bray–Curtis dissimilarity as different dietary groups formed distinct clusters (Figure 5E). Post hoc analysis using ANCOM-BC showed that the only significant change at the phylum level in IUGR-RC compared to CON was decreased Deferribacteres. At the genus level, *Turicibacter*, unclassified Erysipelotrichaceae, *Prevotella* and *Mucispirillium* decreased significantly in IUGR-RC. Groups fed a high fat and high fructose diet either starting prenatally or postnatally after intrauterine caloric restriction showed major differences in bacterial composition or abundance compared to controls. Unlike IUGR-RC, we noticed significant increased abundance of phyla Proteobacteria, Bacteroidetes and Actinobacteria in HFhf group whereas IUGR-HFhf group showed a significant increase in Verrucomicrobia along with Proteobacteria, Actinobacteria but not Bacteroidetes. However, in both groups significant reduction of phylum Deferribacteres was observed. At the genus level, significantly increased abundance was observed for *Lactococcus*, *Blautia*, and *Dorea* in both HFhf and IUGR-HFhf groups. However, only IUGR-HFhf showed abundance of *Akkermansia*, Clostridiaceae, Enterobacteriaceae and *Morganella*. Furthermore, we noticed decreased abundance of Lactobacillus in the HFhf group only (Figure 5F–H).

### 3.3. Effects of Early Dietary Influence on Adult Gut Bacteria

To investigate the long-term effects of early life dietary mismatch, we assessed microbial alpha and beta diversity at 180 days of age using different scoring indexes. Our study revealed significantly decreased alpha diversity for all three experimental groups compared to CON. IUGR-RC group demonstrated the lowest mean alpha diversity for both observed and Shannon index. Identical trends were observed with Phylogenetic diversity as well. However, Simpson index showed significant decline with the IUGR-RC group only (Figure 6A–D). Furthermore, Principal Coordinates Analysis (PCoA) plots of Bray–Curtis dissimilarity showed a distinct separation of clusters based on diet (*p* = 0.001, PERMANOVA). Both CON and IUGR-RC groups form distinct clusters whereas samples from IUGR-HFhf and HFhf have some overlap (Figure 6E). Results obtained from post hoc analysis of composition with bias correction (ANCOM-BC) revealed increased abundance of Firmicutes in all three experimental groups. Simultaneously, we observed an increase in abundance of phylum Verrucomicrobia and Proteobacteria in the IUGR-HFhf and HFhf groups (Figure 6F–H). In line with our findings at phylum level we have seen positive enrichment of a large number of genus among HFhf fed groups, especially in HFhf alone. We have noticed significant abundance *Coprobacillus, Holdemania, Dorea, rc4-4* and *Anaerotruncus* among both HFhf and IUGR-HFhf groups but not in IUGR-RC group. Both *Holdemania, Dorea* are associated with liver cirrhosis [26] *rc4-4* is known as obesity associated bacterium [27]. *Anaerotruncus* sp. are succinate producing bacterium and are reported to be isolated from the stool of obese patients after bariatric surgery [28]. 

Besides bacterial diversity or richness, the ratio between various major taxa can be an indicator of microbial shifts with a strong influence on host metabolism. To this end, we have examined the ratio of Firmicutes to Bacteroidetes, which has been previously linked to obesity. The ratio decreased with administration of HFhf diet and increased with age as adult offspring from all dietary groups had higher Firmicutes/Bacteroidetes ratio (Figure 7A,B). Further analysis was carried out to evaluate the ratio between LPS producing Gammaproteobacteria to SCFA producing Clostridias, which demonstrated an increased ratio in the late postnatal period at PN21 among the two HFhf dietary groups (Figure 7C,D).

Finally, sex-specificity (male versus female at PN21 and 180d) and diurnal variations (between ZT4 and ZT16 at PN21 and 180d) were assessed in the different dietary groups and found to demonstrate no differences; Appendix A).

## 4. Discussion

In our present study we demonstrated the role of prenatal to postnatal dietary mismatch in the IUGR offspring upon the composition of gut microbiota through the life course. Our prenatal calorie restriction rat model has been previously validated and again noted to successfully induce intra-uterine growth restriction in the offspring [29]. In addition, we have previously demonstrated that the chronicity of prenatal caloric restriction results in uteroplacental insufficiency with a reduction in uterine blood flow [30]. The IUGR litters in this study had significantly more pups per litter and demonstrated a higher female to male ratio per litter compared to the CON group and even more so when compared to the HFhf group. It is also known that a high fat diet during pregnancy leads to vascular abnormalities in the placenta [31,32] which in turn compromises the intra-uterine growth of the developing fetus resulting in growth restriction as well. Previous studies have shown that diet plays a pivotal role in the composition of the intestinal microbiome [6,31]. Our present study brought focus specifically to the effects of the mother’s diet during pregnancy and lactation upon the intestinal microbiome of the offspring postnatally. 

Earlier studies showed abundance of Verrucomicrobia in the state of malnutrition [33,34], and it is not surprising that they were more abundant in our PN2 IUGR group. However, the overrepresentation of Verrucomicrobia in the IUGR-HFhf offspring group even at PN21 suggests longer lasting effects of the in utero dietary restriction with further exaggeration when exposed postnatally to a HFhf diet. This is because in contrast, no significant enrichment of Verrucomicrobia emerged in the PN21 IUGR-RC group. The calorie restriction endured prenatally by the offspring appears to have long-term effects upon the intestinal microbiome even later in life during the adult stage, regardless of the dietary exposure, and potentially contributes to the phenotypic differences observed in the adult IUGR offspring. Combining our previous observations [35] with some of our present observations, the IUGR-RC and IUGR-HFhf adult offspring demonstrated central adiposity, glucose intolerance and hyperlipidemia. Corresponding to these findings, the groups that were raised on a HFhf diet either prenatally and postnatally (HFhf) or postnatally alone (IUGR-HFhf) displayed more Proteobacteria. This observation is consistent with a Western diet being proinflammatory, causing metabolic disorders, and cardiovascular disease later in life. While obesity did not emerge during the postnatal stages, at 180d of age, the IUGR-HFhf adults (particularly the males but also the females) were observed to be heavier than the CON and IUGR-RC groups, and trended higher than the HFhf group (Appendix A). 

The two largest phyla in the gut microbiome were observed to be Firmicutes and Bacteroidetes. The Firmicutes/Bacteroidetes ratio has been associated with a few pathological conditions including obesity, disruption of metabolic homeostasis and colorectal cancer [36,37]. Obesity has been specifically associated with a greater abundance of Firmicutes and a drop in Bacteroidetes, i.e.**,** increase in the ratio. However, some research has shown no change or even an increase in Bacteroidetes with obesity. Simultaneously, a high-fiber diet can increase the abundance of Firmicutes and reduce the abundance of Bacteroides and consequently increase the concentration of short-chain fatty acids (SCFAs) in the intestine, inhibiting the development of colorectal cancer. The Firmicutes/Bacteroidetes ratio undergoes an increase from birth to adulthood and is further altered with advancing age. This ratio appears applicable in highlighting variations between infants, adults, and the elderly. It can be linked to overall changes in bacterial profiles at different stages of life [38]. We have observed a decrease in Firmicutes/Bacteroidetes ratio with the administration of HFhf diet both prenatally and/or postnatally alone, while the ratio increased in all four groups with age.

High abundance of most of the bacteria of phylum Firmicutes is an indicator of unhealthy lifestyles, such as a high-fat diet [39], abnormal energy balance [40,41], culminating in obesity [42]. However, some bacteria from this phylum are short-chain-fatty-acid (SCFA) producers, such as acetate, butyrate-producing species, that seem to have a beneficial impact on health. Clostridiales, a class of the phyla Firmicutes, are anaerobes, and some of the members are involved in Short Chain Fatty Acid (SCFA) production, mainly butyrate, one of the main energy sources for colonocytes and helps maintain the mucous layer lining the intestinal epithelium. We have observed significant enhancement of some of the SCFA producing Clostridiales, i.e., *Butyrivibrio, Roseburia, Blautia* along with other short chain fatty acid producing *Eubacterium* spp. in both HFhf and IUGR-HFhf groups. Simultaneously, we have noticed significant abundance of Gamma-proteobacteria, a LPS producing bacteria of the gut. Gamma-proteobacteria, a class of the phyla Proteobacteriae, is proinflammatory and produces lipopolysaccharides (LPS), being toxic upon release into the bloodstream, and suggested to play a role in the pathophysiology of necrotizing enterocolitis. Gammaproteobacteria (Genus: *Morganella, Proteus*) were also significantly increased in both the HFhf fed groups. To this end we measured the ratio of the Gamma-proteobacteria/Clostridiales ratio (G/C) from all experimental groups. The over-abundance of gamma-proteobacteria and decreased amounts of Clostridiales in HFhf or IUGR-HFhf diet groups are consistent with the development of dysbiosis at PN21 in our postnatal fed HFhf diet groups. Furthermore, it speaks to a heightened inflammatory state in those that were raised on a diet high in fat and fructose. 

Our results in the IUGR-RC group being no different from CON despite developing central adiposity and glucose intolerance, suggests that the state of IUGR alone may not be responsible for the gut dysbiosis encountered through the life course of the offspring, whether male or female, but the superimposition of a high fat/high fructose hypercaloric diet either prenatally or postnatally results in significant disruption of the gut microbiome perhaps contributing towards the ultimate phenotype and setting the offspring up for immediate postnatal complications and long term consequences throughout the life course. Our present study highlights the consequences of a nutritional mismatch particularly when a high fat with high fructose diet is introduced early in life causing dysbiosis. 

The strength of our study is the longitudinal nature from prenatal (dams) to soon after birth all the way until 6 months of life of the male and female offspring during the light and dark phases (also seen in Appendix A). Another strength is that both the control and IUGR offspring were exposed to either a regular chow diet or a high fat and high fructose diet to allow for simultaneous comparison between the IUGR and CON, and between the two different dietary regimens. This study design led to confirming the observation that IUGR pups in the immediate postnatal period (PN2) exhibit a distinctly different microbiome when compared to controls, supporting the presence of dysbiosis. The postnatal introduction of differing diets, namely a high carbohydrate regular chow diet versus a high fat and high fructose diet while allowing age-matched comparisons, supported the high fat and high fructose diet acting as a second postnatal hit that triggered further dysbiosis resembling a dietary effect (at PN21 and 180d). The limitation of our study is that the observations were associative and lacked testing of a mechanism as can be undertaken in vivo by conducting fecal microbial transplantation experiments. Such experiments can provide a cause-and-effect paradigm that connects the observed microbial changes with the observed metabolic and phenotypic presentations previously described in the IUGR-HFhf versus the IUGR-RC or control adult offspring.

## 5. Conclusions

The long-term goal of this research is to identify areas of possible interventions during early life to promote development of a healthy microbiome in the neonate which may reduce the incidence of immediate life-threatening diseases (e.g., NEC) and late onset sepsis as well as future chronic, non-communicable diseases such as obesity and diabetes that exert such a severe cost on society. We conclude that the IUGR offspring combined with a HFhf diet develop dysbiosis which creates a particular high risk for developing detrimental health complications later in life.

## Figures and Tables

**Figure 1 nutrients-15-00217-f001:**
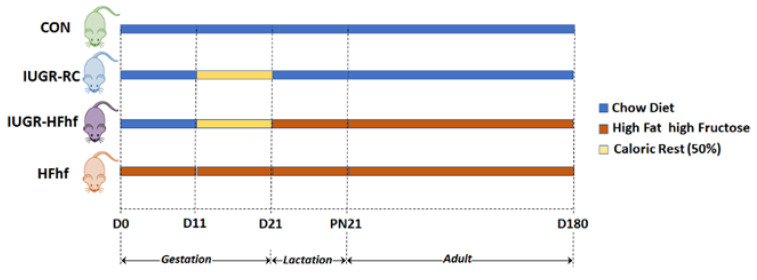
Scheme of the study design, depicting all dietary groups and duration of caloric restriction. For PN2 and PN21 samples 7–10 male and female rats were assigned to each group. For D180 samples 12–16 male and female rats were assigned for each group. D = day of life with D0 equaling the day of conception, PN = postnatal, CON = control, IUGR-RC = Intra-uterine growth restriction on a regular chow diet, HFhf = high fat and high fructose diet, IUGR-HFhf = Intra-uterine growth restriction on a high fat high fructose diet. Key: Caloric Rest = Caloric restriction.

**Figure 2 nutrients-15-00217-f002:**
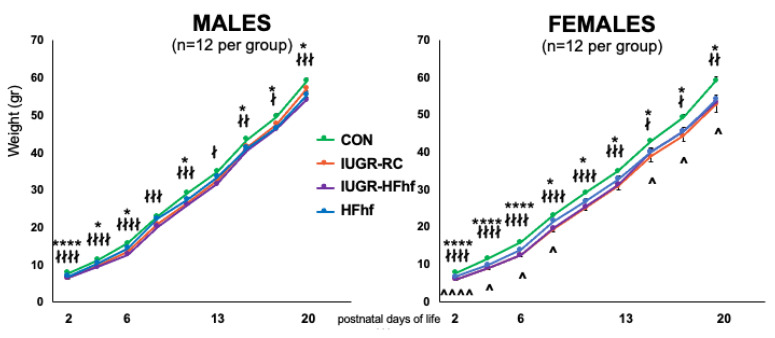
Postnatal Body weights in males and females of the four experimental groups. CON = control, IUGR-RC = Intra-uterine growth restriction on a regular chow diet, HFhf = high fat and high fructose diet, IUGR-HFhf = Intra-uterine growth restriction on a high fat high fructose diet. n = 12 litters per experimental group were examined. * compares HFhf versus CON, ∤ IUGR-RC versus CON, ∧ compares males versus females, * or ∤ *p* < 0.05, ∤∤ *p* < 0.01, ∤∤∤ *p* < 0.005, **** or ∤∤∤∤ *p* < 0.0001. ^ *p* < 0.05, ^^^^ *p* < 0.001.

**Figure 3 nutrients-15-00217-f003:**
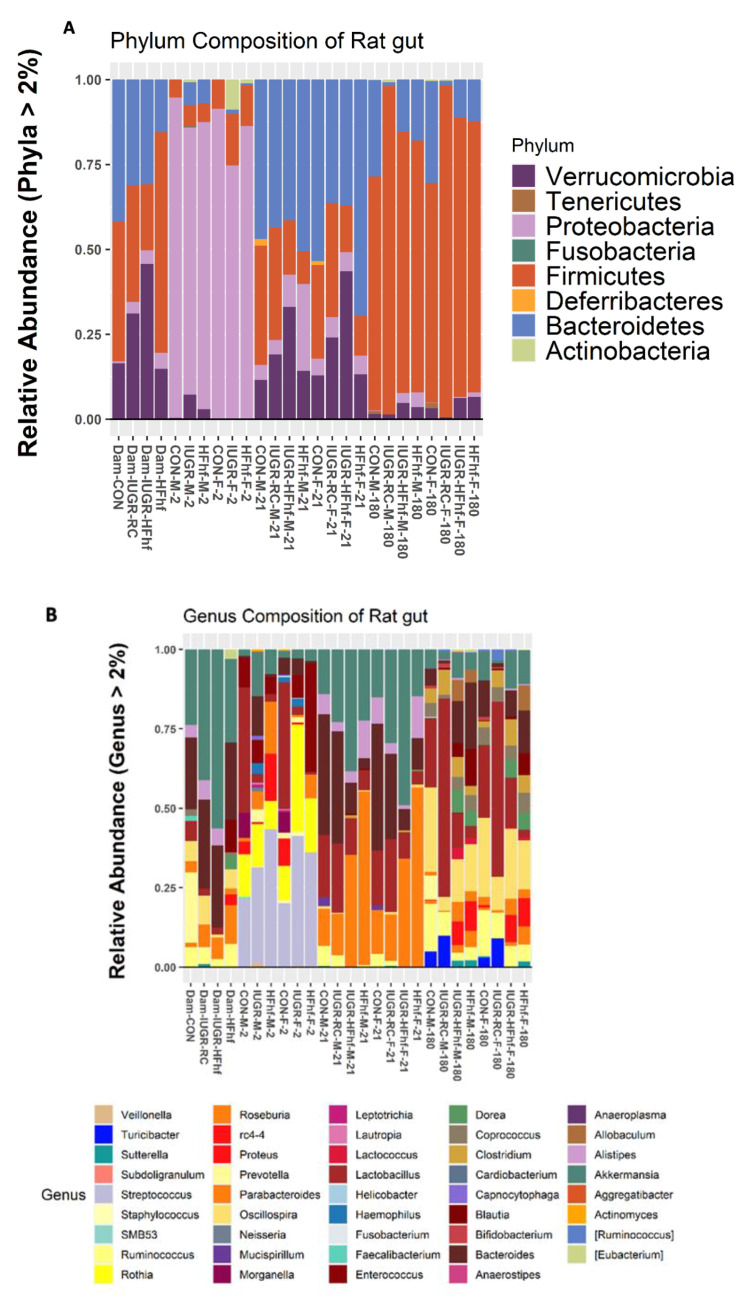
Taxonomic profiles of fecal microbiota by 16S rRNA gene sequencing at the Phylum (**A**) and Genus (**B**) levels.

**Figure 4 nutrients-15-00217-f004:**
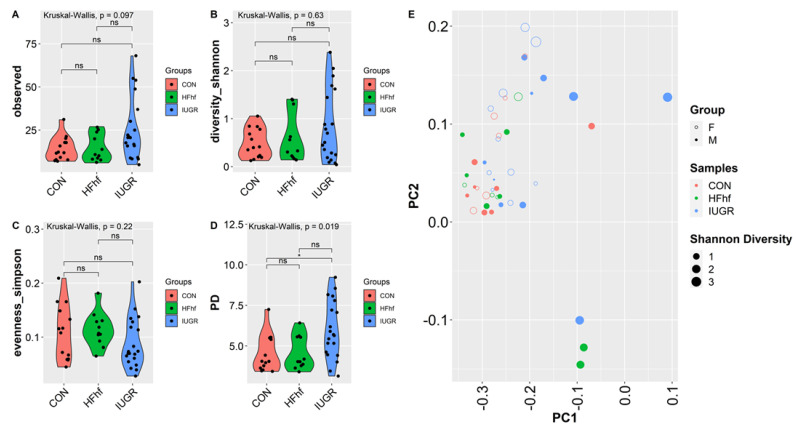
Effects of dietary influences on gut bacterial diversity at PN2: Violin plots showing gut microbiota alpha diversity at postnatal day 2 (PN2) (**A**–**D**) with four different matrices. Kruskal–Wallis *p*-values are denoted for each comparison with post hoc Dunn test for comparison between different dietary groups. Significance levels are marked as * *p* < 0.05. n.s., not statistically significant. (**E**) Principal coordinates analysis of Bray–Curtis distance based on 16S rRNA gene profiling of feces collected from early postnatal stage (PN2) of different dietary groups. (**F**,**G**) Evaluation of the relative abundance of microbial taxa in different dominant diet categories using differential analysis between IUGR and CON (**F**), and between HFhf and CON (**G**). The bars represent log2 fold change. Only microbial taxa with significant fold changes (*p* < 0.05) at the phylum (**left**) and genus (**right**) levels are shown. Members of microbial families that are not classified at the genus level are also included in the right panels.

**Figure 5 nutrients-15-00217-f005:**
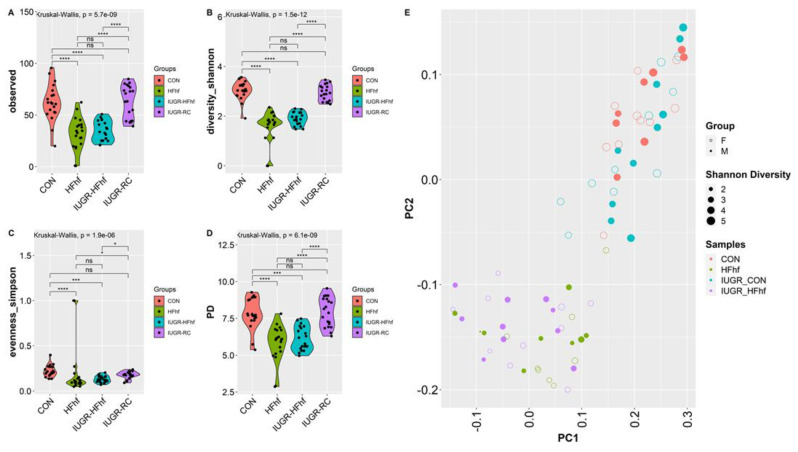
Violin plots showing gut microbiota alpha diversity at postnatal day 21 (PN21) (**A**–**D**) with four different matrices. Kruskal–Wallis *p*-values are denoted for each comparison with post hoc Dunn test for comparison between different dietary groups. Significance levels are marked as * *p* < 0.05, *** *p* < 0.001, **** *p* < 0.0001. n.s., not statistically significant. (**E**) Principal coordinates analysis of Bray–Curtis distance based on 16S rDNA profiling of feces collected from late postnatal stage (PN21) of different dietary groups. (**F**–**H**) Evaluation of the relative abundance of microbial taxa in different dominant diet categories using differential analysis between IUGR and CON (**F**), between HFhf and CON (**G**) and between IUGR-HFhf and CON (**H**). The bars represent log2 fold change. Only microbial taxa with significant fold changes (*p* < 0.05) at the phylum (**left**) and genus (**right**) levels are shown.

**Figure 6 nutrients-15-00217-f006:**
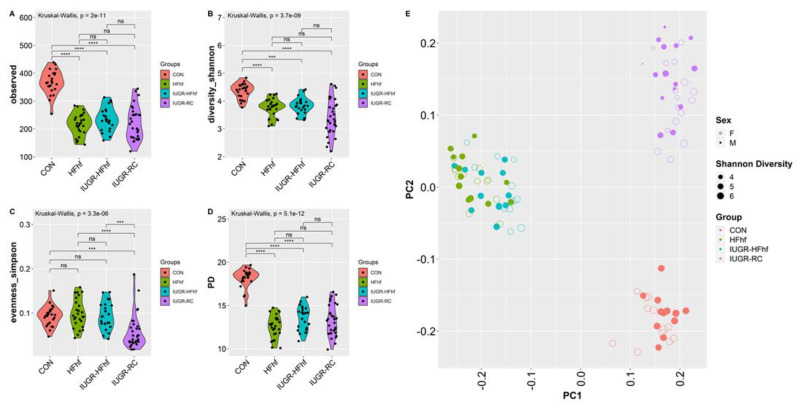
Violin plots showing gut microbial alpha diversity in the adult offspring (180d) (**A**–**D**) with four different metrics. Kruskal–Wallis *p*-values are denoted for each comparison with post hoc Dunn test for comparison between different dietary groups. Significance levels are marked as *** *p* < 0.001, **** *p* < 0.0001. n.s., not statistically significant; (**E**) Principal coordinates analysis of Bray–Curtis dissimilarity based on 16S rRNA gene profiling of feces collected from the adult offspring (180d) of different dietary groups. (**F**–**H**) Evaluation of the relative abundance of microbial taxa in different dominant diet categories using differential analysis between IUGR and CON (F), between HFhf and CON (**G**) and between IUGR-HFhf and CON (**H**). The bars represent log2 fold change. Only microbial taxa with significant fold changes (*p* < 0.05) at the phylum (**left**) and genus (**right**) levels are shown.

**Figure 7 nutrients-15-00217-f007:**
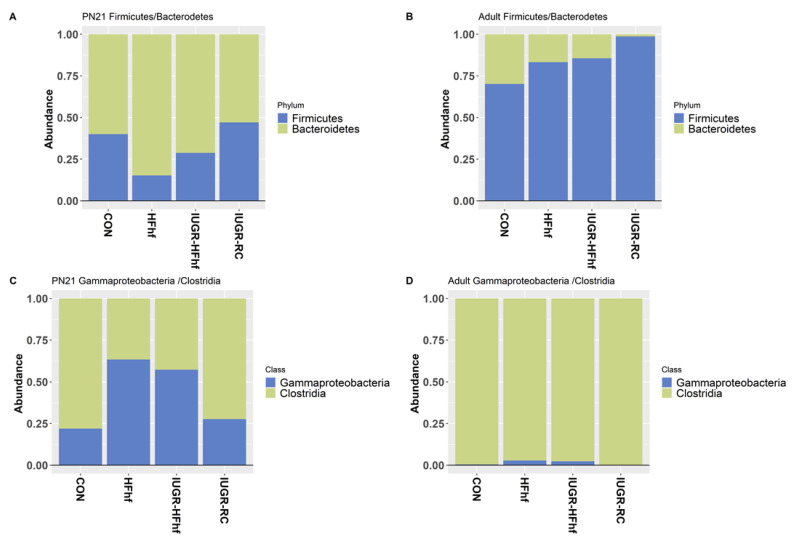
Stacked bar plots showing the ratio of Firmicutes to Bacteroidetes (**A**,**B**) and Gammaproteobacteria to Clostridia (**C**,**D**) among different dietary groups.

## Data Availability

The data discussed in this publication have been deposited in NCBI’s Gene Expression Omnibus [43] and are accessible through GEO Series accession number GSE220146.

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
