# Peer review of "Variation in the Early Life and Adult Intestinal Microbiome of Intra-Uterine Growth Restricted Rat Offspring Exposed to a High Fat and Fructose Diet"

_nutrients, 2023, doi:10.3390/nu15010217_

Round 1

Reviewer 1 Report

The manuscript revealed the effects of prenatal undernutrition and prenatal and postnatal high-fat-high-fructose diet on the gut microbiota. Furthermore, the authors suggest that maternal calorie restriction during pregnancy have long-term effects on intestinal microbiome of the offspring and dysbiosis by a mismatch between IUGR and high-fat-high-fructose diet leads to the increased risk of health problems in the offspring. This paper is well-written and interesting, however, I have some concerns to be addressed.

1.      It is difficult to read some letters in the figures because the size of the letters is too small. Please enlarge the letter size.

2.      Line 200, page 6; Clostridiales was not reduced in the IUGR compared to CON in Figure 3F though the author mentioned that Clostridium was reduced. Please explain the details.

3.      Line 233-234, page 8; Actinobacteria, but not Bacteroides, increased in IUGR-HFhf group in Figure 4H. Please correct it.

4.      Line 332 and 340, page 13; The references containing “Unpublished data” should be added in the Reference section, or those data should be presented in the manuscript or supplementary materials. Especially, the fact that the IUGR-HFhf adults were heavier than the HFhf group may strengthen the author’s hypothesis that difference in microbiota between HFhf and IUGR-HFhf adults is involved in the metabolic state of the rats.

Author Response

Comment: The manuscript revealed the effects of prenatal undernutrition and prenatal and postnatal high-fat-high-fructose diet on the gut microbiota. Furthermore, the authors suggest that maternal calorie restriction during pregnancy have long-term effects on intestinal microbiome of the offspring and dysbiosis by a mismatch between IUGR and high-fat-high-fructose diet leads to the increased risk of health problems in the offspring. This paper is well-written and interesting, however, I have some concerns to be addressed.

Response: Thank you for finding our paper well-written and interesting, we have addressed the concerns below:

  1. It is difficult to read some letters in the figures because the size of the letters is too small. Please enlarge the letter size.

Response: In response to this concern, we have enlarged the font size in all figures, especially Figures 3-6

  1. Line 200, page 6; Clostridiales was not reduced in the IUGR compared to CON in Figure 3F though the author mentioned that Clostridium was reduced. Please explain the details.

Response: We have rectified this error made by us, by stating that it is increased.

  1. Line 233-234, page 8; Actinobacteria, but not Bacteroides, increased in IUGR-HFhf group in Figure 4H. Please correct it.

Response: We have made this correction.

  1. Line 332 and 340, page 13; The references containing “Unpublished data” should be added in the Reference section, or those data should be presented in the manuscript or supplementary materials. Especially, the fact that the IUGR-HFhf adults were heavier than the HFhf group may strengthen the author’s hypothesis that difference in microbiota between HFhf and IUGR-HFhf adults is involved in the metabolic state of the rats.

Response: We have uploaded a plot demonstrating the differences in body weight between all 6 month old male and female rat offspring from the four experimental groups as Supplementary Figure 1. All other reference to unpublished data has been deleted.

Reviewer 2 Report

This is an interesting manuscript describing an in vivo study on the variation in the early life and adult intestinal microbiome of Intra-Uterine Growth Restricted rat offspring exposed to a  high fat and fructose diet. I have the following comments and suggestions:

In the Method section, please include the number of rats in each group.

In the Method section, please include a paragraph on the statistical analyses

In lines 152-155, please include numbers for these findings

Figure 1, please include the meaning for abbreviations

Figures 3-4: I think there are a lot plots in these panels. Please consider to use two separate panels for each Figure.

In general, please improve the resolution and readability of each figure.

Please include a limitation section

Author Response

Comment: This is an interesting manuscript describing an in vivo study on the variation in the early life and adult intestinal microbiome of Intra-Uterine Growth Restricted rat offspring exposed to a  high fat and fructose diet. I have the following comments and suggestions:

Response: Thank you for describing our manuscript as interesting, we have addressed your suggestions below:

  1. In the Method section, please include the number of rats in each group.

Response: We have included the total number in the methods section, divided equally between the four experimental groups.

  1. In the Method section, please include a paragraph on the statistical analyses

Response: We have added a paragraph on the statistical analyses as suggested by this reviewer.

  1. In lines 152-155, please include numbers for these findings

Response: We have included a Supplementary table 1 describing the number of pups and the litter size in each experimental group at PN2.

.4.    Figure 1, please include the meaning for abbreviations

Response: We have added the expansion for all abbreviations in the legend for figure 1.

  1. Figures 3-4: I think there are a lot of plots in these panels. Please consider to use two separate panels for each Figure.

Response: We have collated all the separate panels of figures 3 and 4 into two major panels each.

  1. In general, please improve the resolution and readability of each figure.

Response: We have increased the font size in all our figures. The resolution of all our figures are 600 dpi

  1. Please include a limitation section

Response: We have included a strengths and limitations section to our discussion.